# MAC-CAFE: Multi-actor, Centralized Critic Architecture for Feedback-driven Editing

## Abstract

Large Language Models (LLMs) often generate incorrect or outdated information, especially in low-resource settings or when dealing with private data. To address this, Retrieval-Augmented Generation (RAG) uses external knowledge bases (KBs), but these can also suffer from inaccuracies. We introduce MAC-CAFE, a novel **M**ulti-actor, **C**entralized **C**ritic **A**rchitecture for **F**eedback-driven **E**diting approach that iteratively refines the KB based on expert feedback using a multi-actor, centralized critic reinforcement learning framework. Each document is assigned to an actor, modeled as a ReACT agent, which performs structured edits based on document-specific targeted instructions from a centralized critic. Experimental results show that MAC-CAFE significantly improves KB quality and RAG system performance, enhancing accuracy by up to 8% over baselines.

## 1 Introduction

Large Language Models (LLMs) often produce incorrect or outdated information, particularly in low-resource settings or when handling private data. Even if the information provided is accurate, LLMs can generate hallucinated or imaginary content alongside it (Maynez et al., 2020; Zhou et al., 2021). A promising solution to address these issues is the integration of retrieval components that extract relevant information from external knowledge sources, known as Retrieval-Augmented Generation (RAG) (Chen et al., 2017; Khandelwal et al., 2020; Guu et al., 2020; Izacard et al., 2022; Shi et al., 2023). For clarity, we will refer to these external knowledge sources as Knowledge Bases (KBs). However, KBs themselves can suffer from inaccuracies, incompleteness, or outdated content. To address these challenges, there is growing interest in Knowledge Editing (KE) techniques to enhance LLMs with up-to-date and accurate knowledge.

Advancements in KE have focused on updating the model's parameters (De Cao et al., 2021a; Meng et al., 2022; 2023), adding new parameters to model (Huang et al., 2023; Yu et al., 2024), and holding additional memory (Madaan et al., 2022; Wang et al., 2024a;b). Contrary to approaches that either update model parameters or add new parameters that require white-box access to LLMs, memory-based approaches can work with black-box access to LLMs. In similar line of thought, recently, KE approaches have also focused on refining the KBs themselves (Li et al., 2024). For example, the method proposed in Li et al. (2024) continuously updates KBs with new information, such as the current identity of the British Prime Minister. This approach demonstrates that directly editing the KB is more effective than simply adding new documents, which may coexist with outdated or inaccurate ones. Removing older documents is often not feasible, as only certain sections may be incorrect, while other parts could still provide valuable information for different queries. However, in applications like chatbots or code generation using API documentation, where updated information might not be readily available in document form, expert intervention can be crucial (Ramjee et al., 2024; Afzal et al., 2024). In such cases, expert feedback can be used to directly update the KB with accurate information when the LLM produces erroneous results.

To leverage expert or oracle feedback, we propose MAC-CAFE, a **M**ulti-actor, **C**entralized **C**ritic **A**rchitecture for **F**eedback-driven **E**diting technique. Our contributions are as follows:

1. **Introduction of Feedback-Driven KB Editing**: We present MAC-CAFE, a novel framework that refines the KB using structured edits based on expert feedback. This approach

allows for direct, document-level updates without requiring access to LLM parameters, making it applicable to both white-box and black-box LLMs.

2. **Multi-Actor, Centralized Critic Architecture**: We design a multi-agent reinforcement learning framework where each actor is responsible for a specific document, and a centralized critic coordinates updates based on a global reward signal. This architecture ensures that document-level edits are consistent and contribute to the overall accuracy of the RAG system.

3. **Parameterized Action Space for Document Editing**: We propose a parameterized action space for each document-specific actor, enabling fine-grained control over edits, additions, and deletions within each document. This structured action space allows the actors to perform precise modifications based on expert feedback, resulting in a refined KB that better supports the RAG system.

4. **Definition and Evaluation of KB Characteristics**: We define desirable characteristics for KB refinement, including coherence, completeness, and generalizability, and introduce corresponding metrics to quantitatively assess these properties. These metrics provide a systematic way to measure the effectiveness of KB updates.

5. **Empirical Evaluation and Performance Gains**: We demonstrate that MAC-CAFE significantly improves the accuracy and reliability of the QA system in a variety of settings. Through extensive experiments, we show that incorporating expert feedback into document-level edits leads to a substantial reduction in error rates and enhances the KB's ability to support accurate answer generation.

This paper is organized as follows: Section 2 reviews relevant prior work, while Section 3 presents an illustrative example to introduce and explain our approach. Section 4 details the proposed methodology, and Section 5 outlines the desired characteristics for the edited KB along with metrics for evaluation. Section 6 describes the experimental setup, and finally, Section 7 reports the results.

## 2   RELATED WORK

The MAC-CAFE framework addresses a key limitation of current RAG systems: the inability to dynamically update Knowledge Bases (KBs) without retraining or altering model parameters. Our work draws from research in Retrieval-Augmented Generation (RAG), Continual Learning, Model Editing, and feedback-driven prompt optimization, incorporating insights from Multi-Agent Reinforcement Learning (MARL) to propose an effective solution for KB editing.

**Retrieval Augmented Generation (RAG):** RAG systems enhance LMs by retrieving relevant knowledge from a KB based on the input query and appending it to the context, thereby addressing the limitations of standalone LMs that lack sufficient context and produce inaccurate answers (Chen et al., 2017; Khandelwal et al., 2020; Guu et al., 2020; Izacard et al., 2022; Shi et al., 2023). These systems dynamically construct contexts from unstructured KBs without modifying the LM's internal parameters. MAC-CAFE further enhances RAG systems by refining the KB itself based on feedback, ensuring more accurate and up-to-date information.

**Continual Learning:** Continual Learning (CL) methods address the challenge of updating LMs in non-stationary environments by ensuring that new information is learned without forgetting previously acquired knowledge (Jin et al., 2022; Xu et al., 2023; Padmanabhan et al., 2023; Akyürek et al., 2024). These methods are often computationally intensive and require large-scale retraining, making them less suitable for scenarios requiring frequent updates or minimal computational resources. MAC-CAFE, by contrast, leverages expert feedback to perform direct edits to the KB, avoiding the need for extensive retraining.

**Knowledge Editing:** Knowledge Editing approaches fall into two categories: **Model Editing**, which modifies the LM parameters directly, and **Input Editing**, which updates the knowledge supplied to the model. While Model Editing efficiently alters specific facts using specialized secondary models or altering parameters (De Cao et al., 2021b; Meng et al., 2023), it struggles to ensure consistent updates across contexts (Onoe et al., 2023; Hua et al., 2024). In contrast, Input Editing modifies the KB itself, enabling updates to be reflected in outputs without changing model parameters (Madaan et al., 2022; Wang et al., 2024a;b; Li et al., 2024). MAC-CAFE builds on input editing techniques

by leveraging expert feedback to refine the KB systematically, ensuring more accurate and consistent responses.

**Prompt Optimization:** With the advent of LMs, some recent works approximate gradients in text-based environments using LMs (Pryzant et al., 2023; Wang et al., 2023; Juneja et al., 2024; Gupta et al., 2024) for optimizing task prompts. MAC-CAFE is inspired by these approaches and generates textual reflections, similar to MetaReflection (Gupta et al., 2024) and Shinn et al. (2023), as proxies for gradients. It provides actionable guidance for document updates without the need for differentiable models. Additionally, MAC-CAFE adopts clustering strategies for feedback aggregation from works like UniPrompt (Juneja et al., 2024)- ensuring that actors receive coherent and non-redundant instructions.

**Multi-Agent Reinforcement Learning (MARL):** Multi-agent reinforcement learning (MARL) has been applied to various domains, with early research focusing on tabular methods (Busoniu et al., 2008; Canese et al., 2021; Gronauer & Diepold, 2022) and later expanding to deep learning techniques for high-dimensional inputs (Tampuu et al., 2017; Leibo et al., 2017). Studies have explored independent Q-learning (Tan, 1993), agent communication (Foerster et al., 2016; Das et al., 2017), and centralized training with decentralized execution (Gupta et al., 2017). However, most of these approaches do not address the critical challenge of multi-agent credit assignment. Actor-critic methods have been introduced to overcome this limitation by employing centralized critics with decentralized actors (Foerster et al., 2018; Iqbal & Sha, 2019; Wang et al., 2021; Chen et al., 2023). MAC-CAFE extends such actor-critic framework to operate directly on textual content, using the centralized critic to decompose feedback into actionable textual gradients for each document-specific actor.

In the next section, we provide an example to illustrate the KB editing problem, while also providing an overview of MAC-CAFE.

## 3 EXAMPLE AND OVERVIEW

Figure 1 illustrates our technique applied to the ARKS Pony domain (Su et al., 2024a), where a knowledge base (KB) for the low-resource programming language Pony supports a natural language-to-code task. Due to Pony's rarity, language models often generate code that fails to compile. To address this, we use the Pony compiler as an expert to provide feedback in the form of compile errors.

①  *Evaluating the Knowledge Base State*: We start with an initial KB, including documents like `builtin-array.md`. The system retrieves relevant documents based on the given task (e.g., counting non-inversions in an array) and generates a program, which is evaluated by the compiler, resulting in feedback (e.g., compile errors).

②  *Centralized Feedback Analysis*: We analyze compile errors to generate reflections that explain why the errors occurred. For instance, if the `apply` method in the `Array` class is partial and may raise an error, the reflection suggests adding a `?` to handle potential failures. These reflections are matched to the documents they pertain to, refining the understanding of errors.

③  *Distributing Gradients*: Reflections are generalized into gradients, which summarize modifications needed for each document. For example, the theme might be the partial nature of functions like `apply` and `update`, which need better error handling in the documentation.

④  *Generating Edit Actions*: Gradients are converted into structured edit actions, such as adding or modifying content in specific sections of the documents.

⑤  *Re-evaluation and MCTS Search*: After edits are applied, the KB is re-evaluated, generating new feedback and a reward score. This score guides a Monte Carlo Tree Search (MCTS) to explore different states of the KB, iterating through steps ①-④ to progressively refine the KB and improve the system's overall performance.

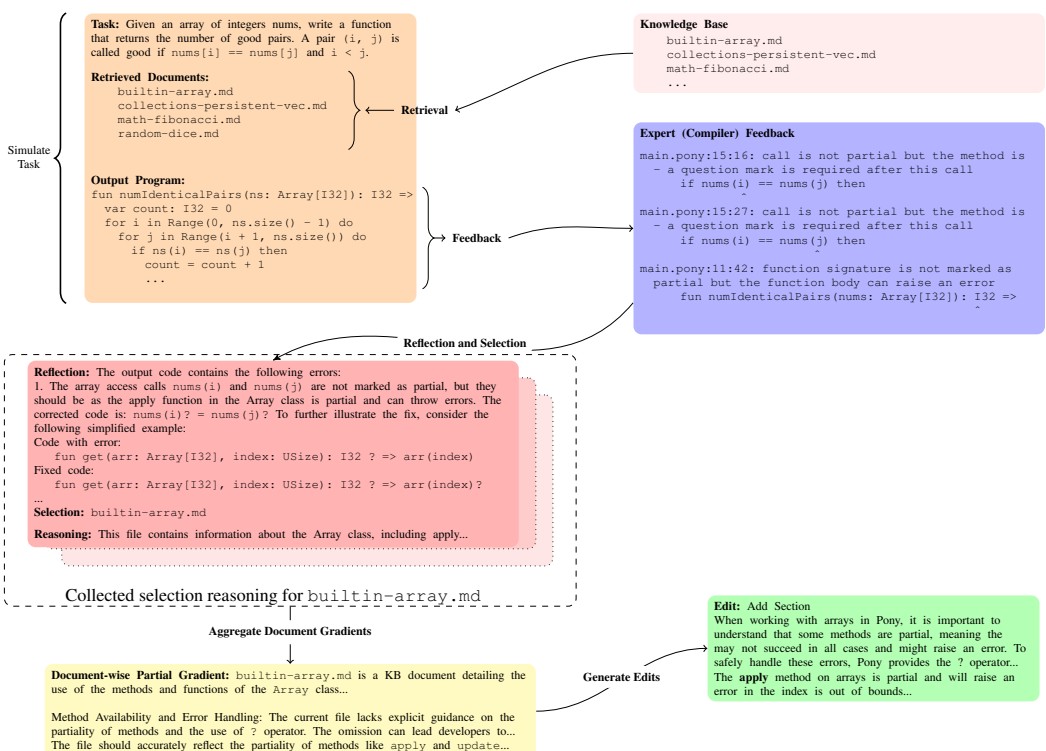

Figure 1: Example of the MAC-CAFE in the ARKS Pony scenario

# 4 METHODOLOGY

We will start by describing a typical Retrieval-Augmented Generation (RAG) system over unstructured Knowledge Bases.

Errors in such systems can arise from multiple components: 1) the LLM $\mathcal{B}$ might fail to reason correctly over the provided information, 2) the retriever $R$ might not select the right set of relevant documents from $\mathcal{K}$, or 3) the knowledge base $\mathcal{K}$ itself might contain incorrect or incomplete information. We assume an expert is monitoring the system, identifying when answers are incorrect, determining which component is at fault, and providing feedback on why the answer is incorrect and what the correct answer must be.

This work focuses on scenarios where incorrect answers result from issues in the Knowledge Base ($\mathcal{K}$). Our goal is to improve $\mathcal{K}$ by addressing mistakes in $\mathcal{K}$ and filling in missing information based on expert feedback, thus enhancing the RAG system's performance on future queries.

## 4.1 PROBLEM FORMULATION

We are provided with a training set $T = \{(q_i, o_i, c_i, f_i)\}_{i=1}^{l}$, where $q_i$ is a user query, $o_i$ is the RAG system's answer, $c_i$ is the correct answer, and $f_i$ is an optional expert feedback on incorrect answers. We also assume access to a scoring function $g$, which compare $o_i$ and $c_i$ to output a score. The objective is to optimize the knowledge base $\mathcal{K}$ to maximize the sum of the scores for all queries in the training set:

$$\mathcal{K}^* = \arg\max_{\mathcal{K}} \frac{1}{|T|} \sum_{(q_i, a_i, c_i, f_i) \in T} g(\mathcal{B}(q_i, \Gamma(q_i, \mathcal{K})), c_i) \tag{1}$$

In the next section, we show how such an objective can be seen as a state search problem.

## 4.2 KNOWLEDGE BASE EDITING AS STATE SEARCH

In our problem setting, the Knowledge Base ($\mathcal{K}$) is defined as a collection of documents $\mathcal{K} = \{D_i\}_{i=1}^n$. We assume each document consists of a number of chunks of text and can be represented as $D_i = [c_{ij}]$. The state $s \in \mathcal{S}$ of the system is represented by the current configuration of the KB, i.e., the content of all documents in $\mathcal{K}$.

Given a query $q_i$ and a set of retrieved documents $\Gamma(q_i, \mathcal{K})$, the LLM $\mathcal{B}$ generates an answer $o_i$. When errors arise due to incomplete or incorrect information in the retrieved documents, our goal is to identify the optimal configuration of $\mathcal{K}$ that improves the accuracy of the system's responses. Thus, we define our state search problem as finding the best state $s^*$ of the KB.

**State Space:** The state space $\mathcal{S}$ encompasses all possible configurations of the KB. Each state $s$ corresponds to a particular set of document contents, represented as: $s = \{D_i\}_{i=1}^n$, where $D_i$ denotes the content of document $i$ and $n$ is the number of documents in $\mathcal{K}$. The state $s$ captures the overall structure and content of the KB at any given point. We set $s_0 = \mathcal{K}$.

**State Transition Function:** The state transition function $\mathcal{T}(s, u)$ defines how the KB changes in response to the action $u$ taken by the agent. Each action contains modifications to one or more documents within the KB, resulting in a new KB configuration. The state transition is formalized as: $s' = \mathcal{T}(s, u)$, where $s'$ is the new state of the KB after applying $u$.

**Action Space:** The action space $\mathcal{A}$ consists of list of diffs $d_i$ corresponding to each document $D_i$. Essentially, $u = [d_i]_{i=1}^{|\mathcal{K}|}$.

**Environment:** We model the environment simply as a "patch" function, that takes the diff generated by the agent and patches the KB to produce the new state.

**Optimization Objective:** Following Equation 1, our objective then is to find the optimal state $s^*$ of the KB that maximizes the overall performance of the RAG system, as measured by a global reward function $R$. The optimization problem is formulated as:

$$s^* = \arg\max_{s \in \mathcal{S}} R(s) = \arg\max_{s \in \mathcal{S}} \frac{1}{|T|} \sum_{(q_i, a_i, c_i, f_i) \in T} g(\mathcal{B}(q_i, \Gamma(q_i, s)), c_i) = a \qquad (2)$$

where $R(s)$ represents the cumulative reward of the KB state $s$, reflecting its ability to support accurate and complete responses for a set of queries.

The reward function $R(s)$ is derived from the expert feedback on the system's generated answers and captures improvements in terms of correctness, coherence, and completeness of the information in the KB. By optimizing for $s^*$, we ensure that the final state of the KB maximizes the overall accuracy and effectiveness of the RAG system, rather than focusing on an intermediate sequence of state transitions.

In summary, the state search formulation defines the problem of finding the optimal state $s^*$ of the KB that maximizes the system's performance. This approach enables us to make targeted, feedback-driven edits to the KB and achieve a refined, high-quality knowledge base that better supports accurate answer generation.

**Monte Carlo Tree Search:** We employ Monte Carlo Tree Search (MCTS) similar to PROMPTAGENT (Wang et al., 2023) to search for the optimal state $s^*$. However, this introduces several challenges: (1) The search space for all possible KB edits is vastly larger than that of standard prompt edits typically explored in the literature (Pryzant et al., 2023; Wang et al., 2023; Juneja et al., 2024; Gupta et al., 2024), making exhaustive search infeasible. (2) Generating actions and subsequent states, as done in methods like PROMPTAGENT, is difficult in the KB editing context since fitting the entire KB into the prompt of a language model is impractical. Despite advancements in handling long contexts (Wang et al., 2020; Kitaev et al., 2020; Press et al., 2022; Su et al., 2024b), these models often struggle to leverage extensive contexts effectively Liu et al. (2024). (3) Finally, the LM would need to output the entire edited KB, which is challenging due to the inherent difficulty LMs face in generating long, coherent outputs (Bai et al., 2024).

To address these challenges, we decouple the KB edits by isolating document-level modifications based on the required updates. Since individual documents can be large, we further break down the edits into manageable sections, enabling a structured editing mechanism that focuses on specific portions of a document at a time. In the next section, we introduce MAC-CAFE, an agent designed to efficiently perform these structured edits based on feedback.

## 4.3 MAC-CAFE

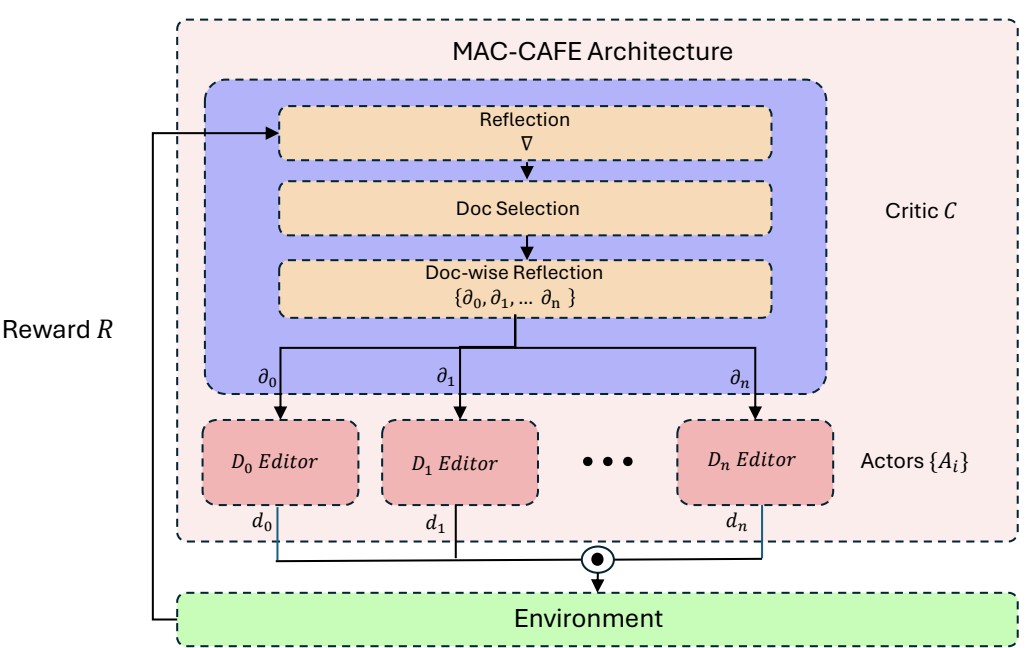

Figure 2: MAC-CAFE Multi-actor, centralized critic architecture: On receiving a reward from the environment, the critic generates a reflection over the failures to calculate the *textual gradient* $\nabla$. The *critic* uses this reflection to select the documents responsible for the error and proceeds to assigns credit to the actors in the form of document-wise reflections. The *actors* then proceed to iteratively edit the documents. All the document-wise edits are then pooled to define the KB edit.

The proposed approach MAC-CAFE is designed to enhance a RAG system by refining the underlying Knowledge Base ($\mathcal{K}$) using expert feedback. Our approach employs a multi-actor, centralized critic architecture, where each actor is responsible for making updates to a specific document within $\mathcal{K}$, and a centralized critic uses global feedback to coordinate these updates. The objective is to iteratively improve $\mathcal{K}$ such that the overall accuracy of the RAG system is maximized.

### 4.3.1 REWARD SIGNAL

For a given query $q_i$ and the generated answer $o_i$, the expert provides feedback $(c_i, f_i)$ that includesa ground truth answer $c_i$ and qualitative expert feedback $f_i$ on any errors. The global reward signal is derived from $c_i$ as per the scoring function $s$ (Refer Equation 2).

### 4.3.2 KB EDITING AGENT

To effectively incorporate expert feedback, we employ a multi-actor, centralized critic architecture.

**Centralized Critic:** The centralized critic, denoted as $C$, is responsible for evaluating the overall performance of the RAG system based on the global reward signal $r$ derived from expert feedback. The critic analyzes the feedback received given the current state $s$ of $\mathcal{K}$. The critic's analysis is then used to provide tailored reflections to each actor, guiding document updates.

The centralized critic aggregates the reward signal across multiple queries to generate a holistic evaluation of $\mathcal{K}$.

$$R(s) = \frac{1}{|T|} \sum_{(q_i, a_i, c_i, f_i) \in T} g(\mathcal{B}(q_i, \Gamma(q_i, s)), c_i) \tag{3}$$

To generate feedback for the documents, the critic needs to take gradient of this reward *with respect to* the documents. This would give us,

$$\partial_j = \frac{\partial R(s)}{\partial D_j} = \frac{1}{|T|} \sum_{(q_i, a_i, c_i, f_i) \in T} \frac{\partial}{\partial D_j} g(\mathcal{B}(q_i, \Gamma(q_i, s)), c_i) \tag{4}$$

Figure 2 illustrates the environmental interaction of the actor-critic model. Following methodologies in prior works (Pryzant et al., 2023; Juneja et al., 2024; Gupta et al., 2024), we use LLMs to generate an overall text gradient $\nabla$ over each failing example. The critic first identifies and select which documents in $\Gamma(q_i, s)$ are responsible for any inaccuracies in $o_i$. Reflections are then generated for these documents based on the correct answer, expert feedback and the text gradient. However, as shown in Equation 4, we need to aggregate these reflections across all queries. Instead of a simple concatenation, we adopt the clustering approach similar to Juneja et al. (2024), producing generalized reflections that effectively capture the core insights from multiple queries. These aggregated reflections can be effectively considered as the partial textual gradient $\partial$ with respect to the document. These partial gradients are provided as feedback to the document-specific actor $A_j$, which then perform the actions to edit the specific documents.

**Actors:** Each document $D_i \in \mathcal{K}$ is managed by a distinct actor, $A_i$, which is modeled as a ReACT agent Yao et al. (2023) responsible for making structured edits to its document. Each actor operates independently, receiving reflections from the centralized critic on how to modify the content of $D_i = [c_{ij}]$. The actors need to only update these chunks as needed. The set of possible actions includes:

- EditChunk: The action is defined as $\text{EditChunk}(j, t_j)$, where $j$ indicates which chunk $c_{ij}$ of $D_i$ to modify, and $t_j$ is the updated content for the chunk.
- AddChunk: The action is defined as $\text{AddChunk}(n_j, t_j)$, where $n_j$ indicates the name of the new chunk, and $t_j$ is the content for the chunk.
- DeleteChunk: The action is defined as $\text{DeleteChunk}(j)$, where $j$ specifies which chunk $c_{ij}$ of $D_i$ to remove.

This parameterized action space allows the actors to perform precise edits within the document, ensuring that the refinement process is both flexible and context-specific. Each actor leverages its local state $s_i$ and the document-specific feedback from the critic to produce a sequence of structured edits, ensuring that modifications are consistent and contribute towards enhancing the document's relevance and completeness.

The ReACT agent utilizes these reflections and iteratively generates a trajectory $t_0 = a_0, a_1, a_2 \cdot a_n$ of edit actions to the document until the errors are resolved or the knowledge gaps are filled. This controlled editing process improves the accuracy of the RAG system by ensuring that the KB contains up-to-date and relevant information. After the completion of the actor runs, we generate the edit diffs for each document $d_i$ and pool them to generate the KB edit action $u = [d_i]_{i=1}^{|\mathcal{K}|}$

However, there might be many ways to edit a KB and we may need to have some desirable characteristics for the edited KB. In the next section, we discuss what those desirable characteristics could be and how we might measure them.

## 5 EVALUATING KNOWLEDGE BASE EDITING QUALITY

A Knowledge Base should be *complete* with respect to a task - it should contain all the information necessary to *assist* the RAG system to solve the task at hand. Given the open-ended nature of tasks

that typical RAG agents are designed for, it is hard to quantify a closed-form metric of *completeness*. That said, an ideal Knowledge Base editing system should at least be able to incorporate as much external feedback as possible.

Further, It will be extremely undesirable for any Knowledge Base to only help the RAG system for a small subset of tasks. Given the tendencies for data-driven techniques to *over-fit* on the train-set distribution, it is important that knowledge base edits are generalizable to unseen examples.

Lastly, given the semantic and textual nature of the Knowledge Base, it is important that the documents in the Knowledge base are coherent and consistent throughout. This not only makes the document interpretable for human consumption, it also help reduce in-context noise during LLM inteference, which has been shown to affect LLM performance (Liu et al., 2024).

# 6 EXPERIMENTAL SETUP

## 6.1 BASELINE

While there has been a rich body of works in the area of knowledge editing and prompt optimization, to the best of our knowledge, MAC-CAFE is the first work targeting the feedback-driven textual Knowledge Base Editing problem. Therefore, to perform a holistic evaluation of MAC-CAFE we implement - PROMPTAGENT-E, an extension of PROMPTAGENT Wang et al. (2023) for the KB editing task. PROMPTAGENT formulates prompt optimization as a strategic planning problem using Monte Carlo Tree Search (MCTS). At a high-level our baseline approach, PROMPTAGENT-E creates separate PROMPTAGENT-style agents to optimize specific document in the KB. To minimize spurious edits in the Knowledge Base, we restrict PROMPTAGENT-E to only optimize documents that were part of the retrievals for more than 2 training sample. After identifying the best nodes for each of the document-wise runs, we put them back in the knowledge base to generate the new version of the KB. In contrast to MAC-CAFE, PROMPTAGENT-E can be seen as a collection of document-wise Independent Actor-Critic models (Foerster et al., 2017). We present in-depth comparisons between PROMPTAGENT-E and MAC-CAFE in Section 7

## 6.2 DATASETS

Knowledge Base Editing can be useful for scenarios where the KB is 1. Incomplete, or 2. Incorrect. We evaluate MAC-CAFE on 5 datasets spanning these different settings.

### 6.2.1 INCOMPLETE KNOWLEDGE BASE

We adapt *two* code generation datasets from ARKS (Su et al., 2024a), namely **ARKS-Pony** and **ARKS-Ring**. The dataset consists of LeetCode problems and their solutions in low-resource languages Pony and Ring respectively. Each datapoint is supplemented with a corresponding language documentation, with execution accuracy as the success metric and execution failures as feedback to the system. Given that these language don't appear promi-

| Dataset | Train | Eval | Test | Documents |
|---|---|---|---|---|
| Pony | 31 | 32 | 45 | 601 |
| Ring | 26 | 27 | 39 | 577 |
| ScipyM | 22 | 22 | 98 | 3921 |
| TensorflowM | 9 | 9 | 26 | 5859 |
| CLARKS News | 30 | 30 | 60 | 138 |

Figure 3: Data splits

nently in LLM pre-training data, the performance of code generation RAG agents on these datasets depends significantly on the quality of the Knowledge Base. However, given that these languages have smaller communities, their documentation isn't as well maintained and often lack critical information. . For the purpose of evaluation on these datasets, we split them into train, eval, test splits as specified in Table 3. To ensure that we have a good representation of failure cases during training, we first execute the RAG pipeline on the entire dataset and divide the failures at random in a 1:1:2 ratio for train, eval and test respectively. All the datapoints with successful execution match are put in the test split. We use the compiler feedback from the executions as the expert feedback to the MAC-CAFE system.

### 6.2.2 INCORRECT KNOWLEDGE BASE

For evaluating under this setting, we leverage the **ARKS-ScipyM** and **ARKS-TensorflowM** datasets from ARKS and the CLARK-news dataset from Erase (Li et al., 2024). The ARKS datasets consist of data science problems sourced from the DS-1000 dataset (Lai et al., 2022), which are to be solved by artificially perturbed versions of scipy and tensorflow libraries respectively, while referring to the original unperturbed documentation. Similar to Pony and Ring, we use the execution accuracy on a test bench as a success metric and use compiler outcome as expert feedback. We also follow a similar approach for data splitting.

While fact retrieval is one of the most popular use cases of RAG systems, evolving nature of information requires us to keep the knowledge bases up to date. To simulate these dynamic factual knowledge updates we use the CLARKS-news dataset from Erase (Li et al., 2024) which contains questions and their respective answers extracted from Wikidata at different timestamps. Each timestamp is characterized by a set of articles that were added in the data at that time. For our evaluation, we pool all the questions whose answers changed for the *first* time at a given timestamp and split them across train, eval and test splits in a 1:1:2 ratio (Table 3).

### 6.3 EVALUATION METRICS

In section 5 we discussed the desirable properties of a Knowledge Base edit. We leverage these properties to design 3 metrics for the KB Editing problem as follows:

**Completeness**: We use the *train set* accuracy to estimate the degree of expert feedback incorporated in the learnt Knowledge Base.

**Generalization**: To estimate the degree of generalization of our Knowledge Base edits, we use the held out *test set* accuracy.

**Coherence**: To quantify the degree of coherence of the KB, we first calculate a document-wise coherence score using G-Eval (Liu et al., 2023) with GPT4-1106-PREVIEW as the judge model. The G-eval prompt assigns a 1-5 score to the *diff* of changes with respect to the original document, checking for thematic similarity of the diff. We pool all the edited documents for a KB edit and average there respective coherence score to define the KB coherence metric.

### 6.4 SYSTEM CONFIGURATIONS

**MCTS parameters:** We use the Upper Confidence bounds applied to Trees (UCT) algorithm for selecting expansion nodes, enabling effective exploration and exploitation of the KB state space. For our experiments, we set a maximum search depth of 3, an expansion width of 3, and a maximum of 5 iterations. The UCT exploration constant is set to 2.5. These parameters were chosen to balance the computational cost and the need for adequate exploration. A depth of 3 ensures that the search can explore sufficient variations in the KB states without unnecessary expansion, while an expansion width of 3 allows a moderate number of candidate states to be evaluated at each step. Similarly, 5 iterations provide enough opportunity to refine the state search, and the UCT constant of 2.5 encourages sufficient exploration in early stages while converging towards high-reward states in later stages. For unstructured data, the documents are chunked after every 50 lines and then edit the chunks.

**RAG System:** For the purpose of our evaluations, we setup a generic RAG system which uses an embedding similarity for semantic retrieval. Additionally, in lines with prior works like (Zhang et al., 2023) for coding related tasks, we use an iterative retrieval setup wherein we first generate a code using naive retrieval and then query the database again with both the question and generated code to improve the quality of retrieval before generating the final result.

**LLM configs:** We use OPENAI-TEXT-EMBEDDING-3-LARGE as the embedding model with dimensions size of 3072 and use cosine similarity as a metric of embedding match for ranking. To account for the 8191 max input limit, we create document chunks of at most 7500 tokens. For the reasoning model, we use GPT4-1106-PREVIEW, with a temperature of 0. Since LLMs are known to perform poorly with longer context input (Liu et al., 2024), we restrict the max token budget for retrievals at 18000 tokens and remove any lower ranked retrieval to fit this token budget.

# 7 RESULTS

| Dataset | Ring | | Pony | | SciPy | | Tensorflow | | CLARK-news | |
|---|---|---|---|---|---|---|---|---|---|---|
| | Acc | $\sigma$ | Acc | $\sigma$ | Acc | $\sigma$ | Acc | $\sigma$ | Acc | $\sigma$ |
| Base KB | 30.77 | 2.09 | 29.99 | 1.57 | 52.04 | 0.00 | 28.88 | 2.18 | 26.27 | 1.20 |
| PROMPTAGENT-E | 33.33 | 2.81 | 32.22 | 1.57 | 53.40 | 3.12 | 47.77 | 3.57 | 28.80 | 2.39 |
| MAC-CAFE | **36.75** | 1.21 | **37.04** | 1.28 | **59.38** | 1.22 | **53.84** | 3.11 | **37.28** | 1.69 |

Table 1: Comparison of Generalization performance of MAC-CAFE and baselines on various datasets

| Dataset | Ring | Pony | SciPy | Tensorflow | CLARK-news |
|---|---|---|---|---|---|
| PROMPTAGENT -E | 4.27 | 3.22 | **33.33** | 33.33 | 11.86 |
| MAC-CAFE | **8.98** | **9.68** | 31.38 | **44.44** | **13.79** |

Table 2: Comparison of Completeness metric for MAC-CAFE and baselines on various datasets

## 7.1 COMPLETENESS AND GENERALIZATION

We observe consistent improvements over the PROMPTAGENT-E baseline in completeness and generalizability scores, with MAC-CAFE achieving approximately 2x performance gains on Ring and Pony datasets. However, feedback incorporation remains limited, likely due to suboptimal retrieval or limited document-query associations hindering generalization. MAC-CAFE also demonstrates higher generalizability and lower variance, attributed to its structured and focused document edits that enhance coherence.

| Dataset | Ring | Pony | SciPy | Tensorflow | CLARK-news |
|---|---|---|---|---|---|
| PROMPTAGENT-E | 4.33 | 1.86 | 2.0 | 4.0 | 1 |
| MAC-CAFE | **4.67** | **4.6** | **4.30** | 4.0 | 1 |

Table 3: Comparison of Coherence metric for MAC-CAFE and baselines on various datasets. Score ranged from 1-5. Higher scores are better

## 7.2 MAC-CAFE MAKES HIGH QUALITY COHERENT EDITS

As seen in Table 3, MAC-CAFE produces edits with a coherence score of 4 or higher for most datasets. For KBs which need long term maintenance (like language and code documentation as seen in the ARKS datasets), MAC-CAFE makes more coherent edits compared to the baseline. This is especially true for long documents as seen in the ARKS Pony dataset. For news-article like dataset like CLARK-news with factual edits. Incoherency is naturally induced when the facts of the article are changed. For instance, an article on the coronation of a king will lose coherency when the article is updated to add information about the coronation of a new king.

# 8 CONCLUSION

We introduced MAC-CAFE, a novel framework for refining Knowledge Bases (KBs) in Retrieval-Augmented Generation (RAG) systems using a multi-actor, centralized critic architecture. MAC-CAFE enables efficient KB updates without retraining or altering model parameters by leveraging feedback-driven structured edits and textual gradients.

Our approach achieved superior performance in preserving knowledge base (KB) coherence, consistency, and completeness, resulting in enhanced RAG system responses. Nonetheless, there remains considerable potential for further advancements. Future work will focus on refining these three metrics to elevate system performance even further.

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
