# A APPENDIX

## A.1 PROMPTS USED IN MAC-CAFE

```
""" There exists a Language Model based software named CodeRAG that automatically does the following
task for a developer: task - task_desc
CodeRAG uses a knowledge base to perform this task: kb_desc
A developer used CodeRAG to perform the task on multiple files, and CodeRAG made some errors on them.
Here is one knowledge base file that was involved in these errors: """ for i, file in
enumerate(kb_files): prompt += f""" File i+1: id: file['id'] content:
n<file>
nfile['content']
n</file>
n""" if "special_notes" in file and file["special_notes"] != "": prompt += f"""
nspecial_notes: file['special_notes']"""
""" The following are the reflections on the errors made by CodeRAG: reflections_str
The reflections show the relationship of the file with the errors made by CodeRAG. If the file is named
"None," it means the information about the error on which the reflection is based does not fully fit any
knowledge base file.
Your task is to use the reflections on the errors made by CodeRAG and provide a generalization on the
issues with the file and how it can be improved to prevent the errors.
You should mention common issues found in the reflections and provide a plan for improving the knowledge
base files to prevent future errors. Use the reflections to suggest additions or changes in the file,
explaining what new content should be added to prevent errors. Before suggesting your plan, give context
on the errors using code snippets and other relevant information from the reflections.
You have a scratchpad to reason and plan your generalization. Your scratchpad is for your use only and
will not be shared with anyone else. The scratchpad is represented by the <scratchpad></scratchpad>
tags.
Your generalization should follow this format: <scratchpad> The contents of the scratchpad </scratchpad>
<generalization> Your generalization for this file </generalization>
You must provide the filled-out scratchpad and generalization in the above format.
General guidelines: 1. Carefully analyze the reflections to understand the errors CodeRAG is making. 2.
"None" is a special file, representing that to fix the error, the information should be in a new file.
"""
```

Figure 4: Generalization Stage Prompt

```
""" Your task is to reflect upon the errors made by CodeRAG based on the user feedback and provide a
reflection on the role of the knowledge base files in the making of those errors.
Your reflection should be very specific to the knowledge base files as these reflections will be used to
improve the knowledge base files to prevent such errors in the future.
There may be other causes for the error, but you should only focus on whether the knowledge base files
could have prevented the error.
You should also provide a way for improving the knowledge base files to prevent the error from happening
again.
You should try and see if there is any error in the information provided by the knowledge base or if the
knowledge base is missing some information that could have prevented the error.
You also have to figure out if the file should be edited or not. That you do through the needs_editing
flag.
You have a scratchpad in which you can reason and plan your reflection. Your scratchpad is for your use
only and will not be shared with anyone else. This scratchpad is represented by the <scratchpad> tags.
Your output should be in the following format:

<scratchpad>
The contents of the scratchpad
</scratchpad>

<reflection>
<File 1>
File: Name of the first file
needs_editing: True/False
Reflection: The reflection for this file
</File 1>

<File 2>
File: Name of the second file
needs_editing: True/False
Reflection: The reflection for this file
</File 2>
...
</reflection>

You have to provide the filled-out scratchpad and the reflection in the above-described formats. You
have to reflect on all the files that were extracted for the code file.
Here are some general guidelines to follow:
    1. You should first analyze the question, the test bench, the feedback, and the output to
       understand the error made by CodeRAG.
    2. Then you should carefully analyze the knowledge base files to see if the theme and the
       contents of any knowledge base file are relevant to the error. Particularly, you should look
       out for files that have a factual error related to the error or are missing some information
       which should have been in the file according to the theme of the file.
        (a) Read the content of the file and understand the theme of the file. The theme of this
            file is of course based on the file ID and the content of the file but you should also
            consider its positioning in the knowledge base. That means you should consider the other
            files that were extracted for the code file and see how this file fits in with them.
            For example, if the file is a very basic general guide to the task with other files
            providing more detailed information, then it would make sense for this file to not have
            detailed information about specific cases.
        (b) See if the file has any information related to the error. Check for relevant keywords
            and how the file might have biased the language model to make the error.
        (c) If the file has information related to the error, see if the information is correct and
            complete. If the information is incorrect or incomplete, the file is responsible for the
            error.
        (d) If it doesnt́ have information related to the error, check if it makes sense for the
            file to have information related to the error. If it doesn't make sense, the file is not
            responsible for the error. When deciding this, check whether the information would be
            better suited in any of the other knowledge base files. If the missing info fits better
            in another file, then deem this file to not be responsible for the error as the missing
            content can be better placed in the other file.
        (e) If the file is responsible for the error, explain the error in your reflection and set
            the needs_editing flag to True. And if the file is not responsible for the error, set the
            needs_editing flag to False.
    3. If none of the files have any error or if you think the content for the error should be in
       a new file, put a file with the name ``None'' in your reflection and for its reflection,
       describe the error and mention why it is not due to the knowledge base files. For the ``None''
       file, the needs_editing flag should always be set to True. The ``None'' file should be placed
       as File n+1 where n is the number of files extracted for the code file.
    4. Choose the least number of files for editing, we want to change as few files as we can for any
       error. For example, if we have 5 knowledge base files, unless very extreme cases, we wouldn't
       want to set the needs_editing flag as True on more than 2 files. Figure out what the most
       relevant files for the error are and focus on them.
    5. When you choose to edit multiple files, you should make sure that their involvements in the
       error are distinct and not overlapping. If they are overlapping, think about whether changing
       one file would be enough to fix the error.
"""
```

Figure 5: Selection Stage Prompt

```
""" There exists a Language Model based software named CodeRAG that automatically does the following
task for a developer:

{test_bench_code}

The test bench code gives a code where a function must be inserted and then it is tested with some

test cases.

CodeRAG then outputted the following code to answer the question:

if task_desc != "":
    prompt += f"""
{task} - {task_desc}
"""
else:
    prompt += f"""
{task}
"""

prompt += f"""

The developer used CodeRAG for a question. The question is as follows:
{query}

In the question, the developer provided the following test bench code:
{test_bench_code}

The test bench code gives a code where a function must be inserted and then it is

tested with some test cases.

CodeRAG then outputted the following code to answer the question:
{output_code}

Based on the above output, the developer gave the following feedback to CodeRAG:
{feedback}

CodeRAG uses a knowledge base to do this task
{kb_desc}

The following files were extracted for this particular code file (the content of

each file is surrounded in <file></file> tags):
"""
for i, instruction in enumerate(instructions):
    prompt += f"""
File {i+1}:
id: "{instruction['id']}"
content: \n<file>\n{instruction['content']}\n</file>\n
"""
    if "special_notes" in instruction and instruction["special_notes"] != "":
        prompt += f"""\nspecial_notes: {instruction['special_notes']}"""

prompt += """
Your task is to reflect upon the errors made by CodeRAG based on the user feedback.
You have to explain in detail the error made by CodeRAG. The reflection should be

very specific to the question, the output code and the feedback.
You should start by explaining the question that CodeRAG was asked to solve before talking about the error.

Your reflection should have relevant code snippets from the output

code which have errors and what should be done to fix them.
You should also add a small code example to demonstrate the error and potential methods to fix it.

You can talk about multiple different methods here to address the error.

You have a scratchpad in which you can reason and plan your reflection.

Your scratchpad is for your use only and will not be shared with anyone else.

Your reflection should be in the following format:
<scratchpad>
The contents of the scratchpad
</scratchpad>
<reflection>
Your reflection
</reflection>
"""

"""
```

Figure 6: Reflection Stage Prompt

```
"""
There exists a Language Model based software named CodeRAG that automatically does the following task
for a developer:
{task} - {task_desc}

CodeRAG was used to perform the task on a specific repository.
The following files in the repository were edited by CodeRAG:
<files>
{files}
</files>

A developer has provided the following feedback on CodeRAG's output:
<feedback>
{feedback}
</feedback>

Your task is to parse this feedback.
You must separate the larger feedback into smaller feedbacks, where each feedback corresponds
to a specific file in the repository.
**DO NOT** change the content of the feedback. Your job is only to split the feedback into file-level
feedbacks, without altering the feedback's content in any way.

You should respond in the following output format:
```json
{
    "task": The task name, like {task},
    "num_files": The number of files mentioned in the feedback,
    "feedback_files": [
        {
            "target_file": The file name that the feedback is about, it should be the

            whole path of the file, leave it empty if you are not sure which file the feedback is about. If
            the file name is not empty,

            it should be one of the files in the repository.

            If the file name does not exist in the repository, put an empty string here,
            "num_feedbacks": The number of feedbacks for this file,
            "feedbacks": [
                {
                    "feedback_tone": The tone of the feedback, can take the values 'positive' or
                    'negative',
                    "target_spans": The spans of the code that the feedback is about.

                    It is a list of json objects with two keys, 'start' and 'end'.

                    Put an empty list if you are not sure which code span the feedback is about,
                    "feedback": Description of the feedback,
                }
            ]
        }
    ]
}
Make sure to include the backticks (```) surrounding the output. They are needed for further parsing of
your output. """
```

Figure 7: Feedback Parsing Prompt