# OpenReview forum: "MAC-CAFE: Multi-actor, Centralized Critic Architecture for Feedback-driven Editing"
_ICLR.cc/2025/Conference — Submitted to ICLR 2025_

### Official Review · Reviewer_eoD8 · 2024-10-28

**Soundness:** 1
**Presentation:** 1
**Contribution:** 2
**Rating:** 1
**Confidence:** 4

**Summary:**

This paper applies ideas from prompt optimization to optimizing KBs for RAG.
The domain considered is text-to-code using documentation as context, focusing on lower-resource coding languages.
The authors take a multi-agent approach, splitting a KB into multiple documents and proposing edits for each document according to a fixed set of edits, based on execution feedback.
After a global document selection stage, each document is edited by a separate editing agent (implemented via REACT prompting and PromptAgent's Monte-Carlo tree search) that proposes a sequence of edits to the document.
Their method, MAC-CAFE, is evaluated on five text-to-code datasets, with two covering cases where the documentation is incomplete and three covering cases where the documentation is incorrect.
The method is evaluated by looking at accuracy before and after editing, as well as coherence metrics.
MAC-CAFE improves accuracy on the test set for all datasets as compared to the base model and a PromptAgent-based baseline.
MAC-CAFE also results in more coherent documentation according to G-Eval.

**Strengths:**

1. **Motivating problem**: The problem of outdated KBs (especially in text-to-code) is interesting and well-motivated, as information in these domains does change consistently. A solution for automatically keeping documentation up-to-date here would indeed be interesting.

2. **Multiple domains**: The authors evaluate the method across multiple domains and datasets and include two different settings (incomplete vs. incorrect info). Their method shows improvements across all domains.

3. **Clear method figure**: Figure 2 is easy to parse and describes the MAC-CAFE method in a way that is easy to understand.

**Weaknesses:**

1. **Limited methodological contribution**: Besides the splitting of the problem into multiple documents, the paper seems like a fairly direct application of PromptAgent with limited technical novelty besides the application to a new domain.

2. **Assumption of error knowledge**: The method uses LLMs to generate code and then use feedback from generated code to update the docs. In lines 194-199, the authors correctly point out that there could be multiple sources of error, including sources that do not stem from incorrect docs. However, the authors then say that they assume errors result from *only* from incorrect docs. It's not clear at all from the writing how -- or whether -- this is enforced, i.e. how the authors ensure that the errors are in fact from errors in the docs rather than the generator's shortcomings. This is especially troubling given that they evaluate on lower-resource coding languages where the model might be worse at generating even with correct docs. If this assumption is enforced, the authors should explain how. If it's not enforced, it's the authors responsibility to convince readers that their benefits come from the system in fact improving the docs in some interpretable way, as opposed to addressing simpler kinds of errors.

3. **Limited results**: the "extensive experiments" mentioned in fact boil down to 2 short paragraphs on the last page of the paper. There are no ablations for the design choices made (e.g. action space, state representation) and no ablations showing the necessity of splitting the task into a multi-actor setup. The authors only evaluate a single LLM. There is no analysis of the resulting KBs after editing.

4. **Unclear baselines and metrics**: It's not clear what the baseline PromptAgent-E is/how it is implemented, why it was chosen, and why it is a fair/relevant baseline. This kind of detail should be brought up in the main paper (which the paper fails to do) and can then be elaborated in supplementary material (which the paper lacks). The authors also do not convincingly argue for the completeness metric measures completeness. The coherence metric is not clearly explained. The metric section is split across 5 and 6.3 in a way that is very hard to follow

5. **Unclear writing and organization**: In addition to the clarity issues above, the rest of the paper also omits a large number of details. To give a few examples:
- The method hinges almost entirely on prompting but the authors do not provide their prompts.
- In Table 3, it's not clear what any of the numbers refer to. They aren't percentages, but they also don't add up to the number of documents.
- Much of the writing could be compressed. There are several sections that conceptually should be subsections/compressed together. At the same time, other parts are overcompressed, e.g. the results section, where Table 3 is unreadably small.

6. **Method described as using gradients**: The method (which is in fact gradient-free) is described in terms of gradients. My feeling is that the authors should make clear that the "gradient" part here is purely a helpful metaphor, as the method does not actually compute any gradients at all. However, from the writing, this point is not brought through clearly.

**Questions:**

One suggestion for measuring effectiveness: you could measure how well MAC-CAFE recovers the existing documentation after a version change. After a major version update (ideally after an LLM's cutoff date) you could give the model the new library along with the old documentation and measure how well you rediscover the new (human-generated gold) docs.

---

### Official Review · Reviewer_wh1j · 2024-11-03

**Soundness:** 2
**Presentation:** 2
**Contribution:** 2
**Rating:** 3
**Confidence:** 4

**Summary:**

This paper proposes MAC-CAFE, a framework to iteratively refine a knowledge based based on expert feedback. They provide experiments on a relatively lesser used coding language, Pony, to simulate a setting where current KBs do not have high amounts of information available. They also provide results on other datasets such as SciPy, Tensorflow, etc.

**Strengths:**

The idea of knowledge-base fixes via edits/additions/deletions is highly relevant today, and MAC-CAFE proposes an interesting and feasible way to approach the same.

**Weaknesses:**

* Section 4.1 that introduces the problem formulation can be written more clearly, especially regarding the following points:
    * Introduce tau(q_i, K) before equation-1 in page 4, to ensure the equation is understandable - currently tau is first introduced in the next subsection.
    * Explain the function g - is g supposed to give a higher score when o_i=c_i?
    * Define o_i=B(qi, tau(qi, K)) [is this the definition?]
* I'd suggest to add a short note describing the ReACT agent (perhaps near line 348) - even though it has been cited, not everyone might have a working knowledge of it.
* Line 348, Actors: I assume a real-life knowledge base would have a huge number of documents - how feasible is it to have a distinct actor model for each document?
* What are the models used for (1) B the LLM, and (2) the actors? I was unable to find it in Section 6. I suggest to add a subsection or paragraph on the same in Section 6.
* I also suggest the authors to add a paragraph each about the baseline PromptAgent and about Monte Carlo Tree Search for more accessibility of the paper.

**Questions:**

* I'd suggest to make the text in Figure 1 (atleast the different stages of your proposed framework) bigger, for ease of reading it!
* Edit suggestion: Line 322, change global reward function from r to R to ensure consistency.

---

### Official Review · Reviewer_wRSu · 2024-11-04

**Soundness:** 1
**Presentation:** 1
**Contribution:** 2
**Rating:** 3
**Confidence:** 3

**Summary:**

This paper addresses the issue of hallucination in large language models (LLMs) within the Retrieval-Augmented Generation (RAG) framework. It introduces MAC-CAFE, a novel multi-agent reinforcement learning approach that iteratively refines external knowledge using expert feedback. Experimental results demonstrate that MAC-CAFE enhances LLM prediction accuracy by 8% compared to baseline methods.

**Strengths:**

1. The focused issue of LLM hallucination is an important problem.

2. The motivation of the study that emphasizes knowledge editing makes sense.

**Weaknesses:**

My major concern lies in two aspects:
1. The rationale behind the design of the proposed method is insufficiently detailed. Although the authors thoroughly describe their implementation, particularly in Section 4, the intuition underlying each component of the design is not clearly described.

2. The result analysis section is quite limited. The authors mainly emphasize the effectiveness of the proposed approach but miss a variety of experiments, such as ablation studies or error analysis, to offer a deeper understanding of its characteristics. For example, which components of the approach are more influential than others? Are there any identifiable patterns in the prediction errors?

Additionally, the paper writing could be improved. For example, I’m a bit confused about the purpose of section 3. If the illustrative example is intended to motivate MAC-CAFE, it might be more effective to condense this description and incorporate it into the Introduction. Doing so would allow for a more detailed and thorough result analysis in the corresponding section.

**Questions:**

Please see the section on Weaknesses.

---

### Official Review · Reviewer_UQrc · 2024-11-06

**Soundness:** 3
**Presentation:** 3
**Contribution:** 3
**Rating:** 6
**Confidence:** 3

**Summary:**

The paper presents a framework for editing and refining information in a knowledge base used in a RAG setup. The framework, MAC-CAFE, has several steps, including a multi-agent updating process where each agent proposes updates for an individual document and updates are aggregated using a critic model. The method improves performance in situations with incomplete or out of date knowledge.

**Strengths:**

S1. This is an important problem, and the paper examines it across several reasonable domains. The method clearly outperforms the baselines presented.

S2. The idea of using multiple agents to control knowledge updates and refinement is a nice one, and seems well-executed here.

S3. I like the idea of defining multiple metrics for knowledge base edits and evaluating along these axes. I think the three chosen are reasonable choices for this task, although I don't have much familiarity with current metrics (if any) used to evaluate this.

**Weaknesses:**

W1. The centralized feedback analysis should be compared with self-reflection / self-refine methodologies.

W2. Managing each document with an agent seems quite computationally expensive-- can you provide some analysis of cost/benefit?

W3. Not a factor in my score-- the contributions listed in the intro don't tie back clearly to sections in the paper; it would be stronger to identify which section each of these contributions is described within. This particular subsection also feels like it may have been LLM-generated; I'm not concerned about that for the rest of the paper, but that section reads a bit poorly.

**Questions:**

Q1. Did you ever consider using some type of knowledge graph instead of a collection of documents?

Q2. Can you show an example of a few steps of the MAC-CAFE process on a few sample documents, given a knowledge update? I think this would help with intuition if this were included as an appendix.

Typos/presentation notes:
- on page 3, a big gap between the end of section 2 and the start of section 3
- Table 2 text is quite large relative to the other tables

---

### Meta-Review · Area_Chair_gEjo · 2024-12-24

**Metareview:**

The paper introduces MAC-CAFE, a novel Multi-actor, Centralized Critic Architecture for Feedback-driven Editing to improve knowledge bases (KBs) used in Retrieval-Augmented Generation (RAG) systems. By employing a multi-agent framework with structured edits guided by a centralized critic, MAC-CAFE refines KBs iteratively using expert feedback.

Strength:
All reviewers agree the problem studied in this paper is quite interesting and has important practical value.

Weakness:
The paper writing needs improvement, such as more clarification about the motivation and rationales behind the designs of the proposed method. And also there is insufficient analysis / ablation experiments to justify the effectiveness of the approach. Also, it is also pointed out by Reviewer eoD8 that this paper has some issues in its error knowledge.

Overall, this paper still needs significant work in its writing and experiments in order to make it publishable.

**Additional Comments On Reviewer Discussion:**

There is no author rebuttal after the review. Therefore, the questions raised by the reviewers are not addressed.

---

### Decision · Program_Chairs · 2025-01-22

Reject